# Interactions of Whey Proteins with Metal Ions

**DOI:** 10.3390/ijms21062156

**Published:** 2020-03-20

**Authors:** Agnieszka Rodzik, Paweł Pomastowski, Gulyaim N. Sagandykova, Bogusław Buszewski

**Affiliations:** 1Department of Environmental Chemistry and Bioanalysis, Faculty of Chemistry, Nicolaus Copernicus University, Gagarina 7, 87-100 Toruń, Poland; agnieszka.rodzik1@gmail.com (A.R.); sagandykova.gulyaim1@gmail.com (G.N.S.); bbusz@chem.umk.pl (B.B.); 2Centre for Modern Interdisciplinary Technologies, Nicolaus Copernicus University, Wileńska 4, 87-100 Toruń, Poland

**Keywords:** whey proteins, metal–protein interactions, food storage, food safety, nutraceuticals, metallocomplexes

## Abstract

Whey proteins tend to interact with metal ions, which have implications in different fields related to human life quality. There are two impacts of such interactions: they can provide opportunities for applications in food and nutraceuticals, but may lead to analytical challenges related to their study and outcomes for food processing, storage, and food interactions. Moreover, interactions of whey proteins with metal ions are complicated, requiring deep understanding, leading to consequences, such as metalloproteins, metallocomplexes, nanoparticles, or aggregates, creating a biologically active system. To understand the phenomena of metal–protein interactions, it is important to develop analytical approaches combined with studies of changes in the biological activity and to analyze the impact of such interactions on different fields. The aim of this review was to discuss chemistry of β-lactoglobulin, α-lactalbumin, and lactotransferrin, their interactions with different metal ions, analytical techniques used to study them and the implications for food and nutraceuticals.

## 1. Introduction

The protein fraction of whey is composed of different proteins, including β-lactoglobulin (β-LG), α-lactalbumin (α-LA), lactoferrin (LTF), small amounts of immunoglobulin (IG), bovine serum albumin (BSA), and lactoperoxidase (LP) [1]. Structures of LTF, β-LG, and α-LA are presented in Figure 1. 

### 1.1. β-Lactoglobulin

β-Lactoglobulin is the major bovine whey protein, accounting for approximately 10% of the total protein in bovine milk and approximately 50% in ruminants [2], but it is not present in human milk. β-LG contains 162 amino acid residues, which form nine antiparallel β-sheets [3]. It belongs to the lipocalin family and has the ability to bind different hydrophobic molecules [4], which can be useful for reducing allergenicity owing to its covalent conjugation to flavonoids because β-LG is one of the major milk allergens responsible for cow milk allergy [5].

β-LG has eleven genetic variants (A, B, C, D, E, F, G, H, W, I, and J). Genetic variants A and B are most common in bovine milk and differ in positions 64 and 118. Two bovine β-lactoglobulins I and J were isolated from bovine milk by isoelectric focusing by Godovac-Zimmermann et al. [4]. Moreover, various variants of β-LG translate into different metal affinities, e.g., to nickel or cobalt complexes [6].

Additionally, β-LG has two disulphide bonds (Cys-106 to Cys-119; Cys-66 to Cys-160) [7] that maintain the structural integrity during hydrolysis and heat treatment and one free cysteine group (Cys-121) as the binding site for d-block metal ions, such as iron (II/III), copper (II), and silver (I) [8]. Due to the disulphide bonds and free sulfhydryl group in its hydrophobic core, β-LG prevents oxidation by capturing reactive oxygen species (ROS) [9]. β-LG may be modified by phosphorylation [10] or glycation [11], which are examples of post-translational modifications (PTMs) of the protein after its translation by proteolytic cuts or by adding a modifying group to one or more amino acids [12]. The molecular weight of β-LG and of other whey proteins is dependent on post-translational modifications (Table 1). It also can be observed that the number of significant figures in the value of molecular weight can be determined by the precision of the analytical method applied for its analysis.

Heating causes changes in a protein’s structure, and subsequently, its properties, thus affecting the quality of food products. De Wit summarized the thermal behavior of β-LG up to 150 °C and concluded that thermal behavior of β-LG is dependent on pH, temperature, time of heating, and concentration [13]. Reversible conformational changes up to 60 °C, which are known as the Tanford transition (negligible between pH 6.5 and 7.8, accounts for 18% at pH 7.0), irreversible denaturation by unfolding and aggregation of monomers between 60 and 70 °C at pH ≥ 7.0 in the presence of OH^-^ ions) have been reported by several authors [13]. In addition, thiols oxidation between 65 and 75 °C, disulphide/thiol exchange reactions prevailing between 75 and 85 °C and induction of larger aggregates by specific non-covalent aggregation, and unfolding of the residual protein structures above 125 °C have been indicated in a number of publications [13]. Liu et al. reported a loss of antioxidant activity of β-lactoglobulin as a result of cross-linking free thiol groups upon heating (100 °C for 2 min) [14]. Wijayanti and co-authors evaluated the effect of lipoic acid in its acidic and reduced forms on heat-induced unfolding of β-LG and obtained results showed that the reduced form was more effective and its effects were similar to N-ethylmaleimide (NEM) and dithio(bis)-p-nitrobenzoate (DTNB) [15]. In contrast to heating, the antioxidant activity of β-lactoglobulin can be enhanced by ultrasound and enzymatic treatment, which modify its secondary structure and strengthen proteolysis [8].

Interestingly, Mercadante et al. [23] reported the ability of bovine β-LG to form dimers and studied the dissociation equilibrium and rate constant over the pH range of 2.5–7.5. The equilibrium constant increased with an increase in |pH-pI|, thus indicating the major role of the hydrophobic effect in the stabilization of the dimer and suggesting that electrostatic repulsion destabilizes the dimer, especially at low pH.

### 1.2. α-Lactalbumin

α-Lactalbumin consists of 123 amino acids, except for rat α-lactalbumin, which contains 17 more amino acids and is an extension of the carboxyl end enriched with proline [24]. α-LA constitutes approximately 22% of the total protein of human milk and approximately 36% of whey protein in human milk, and it constitutes approximately 3.5% of the total protein and approximately 17% of the whey protein in bovine milk [25].

The native α-LA consists of two domains: a large α-helical domain and a small β-sheet domain connected by a calcium-binding loop. α-LA possesses a strong calcium-binding site with residues of Lys79, Asp82, Asp84, Asp87, and Asp88 [26]. Calcium-binding has a significant influence on the molecular stability of LA. Moreover, it is required for the refolding and formation of a native disulphide bond in the reduced, denatured protein [27]. The nuclear magnetic resonance (NMR) and circular dichroism (CD) pH titration studies reported by Kim et al. suggested that critical electrostatic interactions concentrated in the calcium-binding region contribute to the denaturation of the protein by determination of the pK_a_ values of individual functional ionizable groups [28]. When calcium ions dissociate from α-LA at an acidic pH, the protein adopts the molten globule conformation, which has been described as a compact state with a significant degree of secondary structure in the native protein but with a fluctuating tertiary structure [29]. The molten globule has a weakly folded α-helix domain and a domain with a disordered β-sheet domain [30]. The removal of calcium (II) ions resulted in conformational changes, as indicated by spectral (fluorescence and absorbance) changes [31]. Interestingly, the work of Noyelle and co-authors showed that magnesium (II) binding occurred more likely via interactions with the residues belonging to the zinc (II)-binding site in contrast to its expected binding to the calcium (II)-binding site [26]. A study by Wehbi et al. demonstrated that binding of calcium to bovine α-LA increases the resistance of the protein structure to thermal treatment [32].

α-LA is stabilized by four disulphide bonds between the cysteine residues (Cys-6 to Cys-120, Cys-61 to Cys-77, Cys-73 to Cys-91, and Cys-28 to Cys-111) [33]. The active molecular form of α-LA may have various post-translational modifications in contrast to the native form [19]. Moreover, for structural reasons, the α-LA has a metal affinity to ions of s-block elements, such as magnesium (II) and transition metal ions, e.g., zinc (II), which is especially promoted in the reaction with β4-galactosyltransferase, according to immobilized metal-affinity chromatography (IMAC) [34]. In addition, calcium ions increase the stability of α-LA in its native state [35]. Zinc ions may also bind to the calcium-binding site, thus increasing its absorption and bioavailability. In this way, the α-LA complex of zinc can be used as a natural carrier for the supply of zinc in food systems [36]. 

### 1.3. Lactoferrin

Lactoferrin (LTF) is a highly glycosylated protein of the transferrin family [37] that has a molecular weight of approximately 80 kDa, depending on its post-translational modifications [20], [21]. Wei et al. suggested the presence of five N-glycosylated sites of bovine LTF-a (bLTF-a): -Asn-233, -281, -368, -476, and -545 [38]. The degree of glycosylation may vary and thus determines the rate of resistance to proteases or to very low pH [39].

Lactoferrin consists of a single polypeptide chain with approximately 700 amino acids folded into two symmetrical lobes: a N-lobe and C-lobe. These are homologues with respect to each other (33%–41% homology). Each lobe consists of two domains, such as C_1_, C_2_, N_1_, and N_2_ [37]. Both lobes contain approximately 345 residues, and their disposition in each lobe creates an interdomain pocket with a high affinity to iron; the binding is accompanied by synergistic binding of carbonate ions [39]. In more details, in each lobe, a single Fe atom is coordinated by amino acid side chains that are dispersed in each domain and connecting region because of the changes in conformations occur, causing domains to come together. A distorted octahedral coordination sphere is formed by coordinating ligands as carboxylate-O (Asp), two phenolate-O (Tyr), and imidazole-N (His), which is completed by bidentate binding of carbonate or bicarbonate ion. Carbonate is considered as synergistic since its presence is essential for iron binding. The stability constant for iron (III) complex is high (logβ 28 at pH 7.4) [40,41].

LTF can exist in two forms, apo-Lf and holo-Lf, depending on whether it binds iron (III) or not [37]. In addition to iron, LTF is capable of binding other ions, such as aluminum (III), gallium (III), manganese (III), cobalt (III), copper (II), and zinc (II), but with lower affinity [42]. It was reported that LTF releases iron in acidic conditions (pH below 4) [43], and diferric transferrin readily loses iron at pH < 6.7 [44]. It is also very important to consider the iron saturation and concentration at low pH, especially in places of infection and inflammation, where, as a result of metabolic activity of bacteria or stimulated leucocytes, the pH may be lower than 4.5 [45].

## 2. Interaction of Whey Proteins with Metals

Metal ions interact with proteins, thus affecting their biological activity [46]. The evaluation of these changes is of considerable importance because proteins have many functions in the human body and applications in many industries. The following factors are crucial for the assessment of changes induced by interactions with metals: (a) the creation of new binding sites that determine the interactions of protein with other ligands, (b) changes in the protein structure, (c) interacting groups for studying the nature and thus strength of the interaction, and (d) possible protein aggregation. 

The results of metal–protein interactions may include metalloproteins, metallocomplexes, and nanoparticles. Metalloprotein is primarily formed by coordination bonds between metal ions and functional groups of amino acids, for example, carboxyl, of the protein, thus embedding in the protein structure. These protein functional groups form a special binding site in a form of ‘cavity’ that is determined by protein quaternary structure and its biological activity. In addition, metalloprotein can interact with metal ions, forming either metallocomplexes (as first step) or metal/metal oxide nanoparticles. A metallocomplex is an artificial system defined by weak interactions such as electrostatic, hydrogen bonding, Wan der Waals forces, or donor–acceptor bonds, which are stronger than previously mentioned interactions. The binding affinity of metal ions to protein in the metallocomplex can be defined also by inductive (artificial) binding sites and collective strength of weak interactions. In contrast to metalloproteins, the interactions, leading to formation of metallocomplex, occur mostly via sorption or by intraparticle diffusion (modeled by Weber–Morris), while the metal ion of metalloprotein is embedded or ‘buried’ in a protein structure forming a natural system as for, e.g., hemoglobin, transferrin, etc. Interactions of a protein and metal ions with active functional groups of amino acids can be reversible and labile and lead to the formation of nanoparticles as was reported in the study of LTF and silver ions [47]. The formed system consisting of metalloprotein, the formed metallocomplex, and nanoparticles can be considered as a nanocomposite [48]. A graphical representation of the possible results of metal–protein interactions is illustrated in Figure 2. Additionally, it is worthy to mention that statistically, different types of interactions of metal ion and different functional groups of the protein can occur and even simultaneously occur from a theoretical point of view, however the final result is dependent on the conditions of such interaction (temperature, pH, etc.) that determines conformation of the protein, its folding/unfolding and thus formation of binding sites for the metal ion, orientation of functional groups of the protein (steric factor for interaction). This also explains strong coordination bonds of the metal ion and protein in the metalloprotein since many factors can lead to formation of ‘cavities’ as binding sites for metal ion that are quite challenging to reproduce in the artificial system, thus making it possible the synthesis mostly of metallocomplexes, while metalloproteins are formed mostly by a natural way. 

Besides the formation of metallocomplexes and their nanocomposites, metal-induced protein aggregation also occurs, thus causing a loss in biological activity, protein precipitation from solution thus being detrimental for the quality of the product. Chemistry of the metal-induced protein aggregation can be explained by the Derjagin–Landau–Verwey–Overbeek (DLVO) theory of intraparticle interactions, which treats the stability of a biocolloidal system in terms of balance between attractive van der Waals forces and repulsive electrical double-layer forces [49]. The addition of metal ions shifts the attractive forces to increase, thus strengthening the interactions between protein units, causing the formation of large aggregates. An interesting example was shown by Hedberg et al., where the synergistic effects of iron (II/III), chromium (III), and nickel (II) ions were suggested for aggregation of the human serum albumin (HSA) protein. This study also emphasized the importance of considering the safety of metal-based materials upon exposure to the human body and serves as a brilliant example of changes in the biological activity after protein interactions with metals. Changes in the electrostatic forces upon binding were evidenced by changes in electrophoretic mobility [50]. 

Moreover, such aggregation could cause (a) adsorption of proteins on metal surfaces, creating a problem with cleaning, (b) fouling of the filtration membranes by protein, complicating dairy processing by further disruption of membranes for ultra- and microfiltration, which could be explained by the increase in adhesion of proteins on the metal surface after the addition of metal ions. Interesting studies have been carried out to investigate the factors affecting whey proteins fouling. Yang et al. [51] reported that addition of calcium ions was a more prevailing factor for fouling of β-LG rather than its temperature-induced denaturation. Magens et al. showed the influence of surface type on the fouling performance, deposit structure, and composition in terms of the interactions between whey proteins and surface forces. DLVO theory was applied as mentioned earlier for the analysis of particle–surface forces and surface energy [52]. Such studies serve as direct evidence of the importance of studying whey protein–metal interactions and its implications for daily life. 

The current section will be dedicated to interactions between whey proteins and metals, such as copper, zinc, silver, lanthanum, palladium, and ruthenium, as well as to the effects of iron, chromium, nickel, and calcium on the biological activity of whey proteins.

First of all, the major part of metals described in this review are transition metals because the most common inorganic cofactors in biological systems are d-block metals that facilitate various functions of proteins and their complexes [53], thus making the studies of changes in the biological activity of proteins upon their interaction with metals relevant. Moreover, d-block metals have a natural affinity for interacting with proteins owing to the lack of electrons, thus making them able to coordinate to electron-rich moieties in proteins. This makes metalloproteins widespread in nature because metal cofactors are bound to proteins via coordination bonds: it is estimated that more than 50% of all proteins are metalloproteins [46]. Moreover, such metal ions as copper (II), iron (II), manganese (II), and molybdenum (II) have the ability to have the strongest coordination due to their properties such as density and small atomic radius [54]. More importantly, the presence of transition metals is ubiquitous in the environment, and sources and routes of human exposure can greatly vary. In addition, interestingly, metal complexes are applied as drugs for patients with iron-deficient anemia, kidney diseases [55], cancer (complexes of palladium, ruthenium, and platinum [56]), and malnutrition.

Finally, the increased consumption of metal–microelements as supplements to food increases the probability of their interactions with proteins because global demand for protein-rich food has increased with improved living standards [57] along with the prevailing percentage of proteins functions in the human body.

### 2.1. Nature of the Metal–Protein Interaction

The nature of metal–protein interactions provides insights into the type and strength of the interaction as well as involvement of functional groups of a protein. For example, sorption at the surface of the protein, which is characteristic for metallocomplexes, may occur via weak non-covalent interactions that are unstable and reversible. Coordination bonds that allow embedding of a metal into the structure of protein are stronger.

Polypeptide chains of proteins usually coordinate with the metal ion, and side chains with functional groups can act as an additional binding site for metals, including the imidazole group of histidine, carboxyl group of aspartate and glutamate, the phenol ring of tyrosine [58], and nitrogen of lysine and arginine side chains. Hydrogen, electrostatic, and hydrophobic bonds, and van der Waals interactions are significant for the metal–protein interaction, which has a considerable effect on the stabilization of protein structures [58]. Factors affecting the binding of a metal to proteins include the metal properties, such as the valence state, ionic radius, charge-accepting ability, and free metal concentration in the respective biological compartment [59]. However, studies on the nature of metal–protein interactions and characterization of metal binding to proteins are challenging and carried out by instrumental analytical techniques as well as combinations of techniques, as summarized in Table 2. 

Another factor affecting metal–protein interactions is pH. Magyar et al. reported interesting results showing potential pitfalls during metal–protein interactions studies and discussed how pH, temperature, use of different buffers, and the presence of competing ligands affect the K_d_ value [70]. The effect of pH on such interactions can be explained by the protonation state of amino acids of the proteins. Firstly, Asp, Glu, and His respond to pH changes leading to association/dissociation of their complexes with metal ions and they are deprotonated at neutral-alkaline pH that leads to the increase in electrostatic attraction and strengthening of the complex, while in acidic pH protonation occurs and weakens the attractive forces [61,71]. Furthermore, cysteine has a thiol group that has to be deprotonated to be involved in metal coordination as well as tyrosine can be deprotonated to produce a phenolate oxygen donor atom, which, e.g., can be a good ligand for Fe (III) [40]. Interesting, that in the case of the effect of pH on binding affinity of iron to LTF, the carbonate ion is involved, since it is essential for iron binding and this ion is unstable at low pH leading to a release of iron from LTF [40]. Another aspect of the effect of pH is related to conformational changes in a protein structure, e.g., it was reported that β-LG at acidic conditions it caused dimerization that involved changes in the exposed β-strands, but in alkaline conditions the denaturation was observed [72]. 

Tang et al. also showed that glutamate and aspartate should be combined with a nitrogen donor or a sulphur donor to facilitate zinc-binding in peptides or proteins by using isothermal titration calorimetry (ITC). Whey proteins, such as lactoferrin, α-lactalbumin, and β-lactoglobulin showed strong zinc-binding affinities that were similar to each other, even though zinc binding to bovine serum albumin and lactoferrin was exothermic while binding to α-lactalbumin and β-lactoglobulin was slightly endothermic. In addition, authors suggested that zinc binds to the disulphide bonds of oxidized cysteine in LTF and to the thiol group of the cysteine (Cys34) in BSA with significant heat evolution, whereas zinc binds to histidine, aspartate, or glutamate in α-LA and β-LG [60].

Shahraki et al. reported ultraviolet-visible spectroscopy (UV-Vis) results that showed that the interaction of the lanthanum (III)–cysteine complex with β-lactoglobulin and bovine serum albumin induces conformational changes for both proteins. In addition, the lanthanum (III)–cysteine complex strongly quenched the fluorescence of Trp fluorophore in β-LG and BSA in the static quenching mode. Hydrogen bonds and van der Waals forces stabilized the complexes for both proteins [63]. Similar studies on lanthanum–protein interactions were carried out for a lanthanum (III) complex with tryptophan [64] and phenylalanine [65] in relation to human serum albumin. In both cases, spectroscopic techniques indicated the inhibition of protein fluorescence by a static quenching mechanism, whereas data on thermodynamic parameters indicated hydrophobic interactions and hydrogen bonds between the lanthanum (III) complex and protein. In addition, structural studies indicated the conformational changes in proteins in the presence of the lanthanum complex. The lanthanum complex with tryptophan also showed moderate to good antibacterial activity against different bacterial strains. Comparable results were obtained for interactions between human serum albumin and β-lactoglobulin with palladium (II) complexes [66,67]. Spectroscopic studies indicated conformational changes in proteins as a result of the action of the palladium (II) complex. Data on thermodynamic parameters of interaction showed that hydrogen bonds and van der Waals interactions play an important role in HSA/β-LG and palladium (II) complexes associations. In addition, the results of the study showed strong fluorescence quenching of HSA and β-LG by Pd (II) complex via static mechanism [66,67].

### 2.2. Analytical techniques for separation and analysis of whey proteins 

Methods of separation of whey proteins include chromatographic (affinity, anion-, cation-exchange, and reverse-phased), membrane-based (ultrafiltration and microfiltration), and electrophoretic methods. All these methods, except for the electrophoretic, were applied at all stages including isolation, purification, and separation, prior to analysis. Table 3 shows the methods applied for separation of whey proteins for their isolation, analysis, and identification, including real matrixes (different types of cheese and milk). 

Membrane-based methods are very diverse and can be applied for the isolation of whey proteins from real matrices, for the preparation of whey protein concentrate (WPC) and whey protein isolate (WPI), and for separation prior to analysis/detection. 

Electrophoretic techniques include capillary (CE), gel (SDS-PAGE, Native PAGE), and microchip electrophoresis (MCE). The microfluidic “lab-on-a-chip” technique for the separation of proteins has been reported as a high-throughput, automated alternative to conventional SDS-PAGE that allows the separation and quantification of many samples within 30 min. Another advantage is the low sample and material volumes required, which are usually less than 0.5 mL of the total volume chip (10 samples) [76]. In comparison, SDS-PAGE requires several liters of materials (acrylamide solutions, running buffers, and staining/destaining solutions). Thus, Anema S.G. applied this technique for the separation of α-LA and β-LG in different forms and compared with traditional SDS-PAGE and concluded that it is a rapid alternative for separation and quantification of milk proteins [76]. Buffoni et al. used MCE together with LC-ESI-MS to characterize the major proteins from milk of Mediterranean water buffalo [77]. However, the main limitation of the capillary and microfluidic system is the relatively low electro-separation reproducibility owing to the adhesion of proteins to the capillary, the denaturation and sensitivity of the system to changes in pH, and the ionic strength of the buffers. However, Costa et al. indicated that addition of buffers (TPS and SEP) to milk prior to separation showed excellent effects on α-LA and β-LG, whereas separation of caseins was better with the SEP buffer, and the results were comparable to those obtained by SDS-PAGE [75].

Chromatographic techniques for separating whey proteins are also diverse. Separation in this case is based on hydrophobic, ionic, and specific (based on affinity) interactions and consists of adsorption of proteins on a solid (column or membrane) that are eluted with the liquid phase [82]. Ion-exchange chromatography has many advantages over column-packed technology, such as a rapid association rate between the target protein and functional groups, short processing times, ease of scale-up, no heat, and chemical pretreatments or pH changes that could affect the protein structure by altering its properties [80]. Moreover, the column-packed chromatographic technique is expensive and thus, not economically viable suitable for industrial scale-up in the food industry [82]. Anion- [80] and cation-exchange [81] membrane columns were exploited for fractionation of whey proteins from WPC and mozzarella cheese whey, followed by fast protein liquid chromatography (FPLC) and high performance liquid chromatography (HPLC) separation, detection with UV-Vis, and identification with SDS-PAGE, respectively. Doultani et al. [81] found that the one ion exchange system can be used for different purposes depending on the elution buffer. The first step included capturing positively charged whey proteins in the cation exchange column, the second step was the removal of unbound contaminants by rinsing, and the last step involved selective desorption of one or more proteins of interest. One advantage of this study was the use of inexpensive, food-grade buffers within one column to yield high-purity proteins. However, chromatographic approaches in comparison with classical electrophoretic techniques, e.g., two-dimensional gel electrophoresis, still have quite a low resolution power. 

### 2.3. Analytical Techniques for Studies of Interactions of Whey Proteins with Metal Ions 

Instrumental techniques used for the study of the consequences of metal–protein interactions and their potential applications are summarized in Table 4. They include microscopic, spectroscopic, spectrometric, electrophoretic, and even quantum mechanical techniques that provide data regarding the structure, morphology, and chemical composition of metal to protein binding.

#### 2.3.1. Mass Spectrometry

Mass spectrometric methods as highly sensitive techniques that have low detection limits were applied for studying metal–protein interactions by measuring the concentrations of metal ions and protein. Inductively coupled plasma mass spectrometry (ICP-MS) is capable of detecting metal concentrations as it was reported for determination of concentration of the silver (I) ion to study the mechanism of binding of silver to LTF [47]. Acosta et al. used size exclusion chromatography (SEC) in combination with ICP-MS to determine the metals, such as manganese, cobalt, copper, and selenium, present in different whey milk protein fractions of human breast milk (HBM) to detect the elements with appropriate sensitivity and accuracy, whereas MALDI-TOF/TOF-MS and nano-LC–MS/MS were used to analyze the protein fraction composition and quantitative profile [84]. Despite the advantages and fast development of MALDI-TOF-MS protocols for analysis of proteins [96], the gold standard for determination of mass, sequence of proteins, their analysis, and characterization is ESI-MS [97]. Limitations of ESI-MS for metal–protein interactions can include redox reactions that may occur during the ionization, atmospheric pressure that can contribute to oxidation of sensitive species, requirements for high purity of the sample and incompatibility with most commonly used non-organic buffers and salts [98]. Moreover, a drift cell was developed for ion mobility mass spectrometry that allowed one to characterize 14 proteins and protein complexes [99]. In addition, Allen et al. studied the effects of polarity on the structures and charge states of native-like proteins and complexes in the gas phase by ESI-MS and ion-mobility mass spectrometry [100]. Additionally, Lermyte et al. studied metal ion binding to the β-amyloid monomer by native FT-ICR mass spectrometry [101] and effects of transition metals in proteinopathies by ESI-MS [98]. 

#### 2.3.2. Spectroscopic Techniques 

In addition to mass spectrometric techniques, other methods used to understand the mechanism of binding silver to LTF have included Fourier transform infrared spectroscopy (FTIR) and Raman spectroscopy (RS). Both FTIR analysis and Raman spectroscopy have shown significant differences between LTF spectra with the addition of silver from native protein in terms of additional peaks. These techniques are complementary for the study of metal ion interactions with active functional groups of proteins. However, FTIR in comparison with RS is less specific and sensitive owing to the presence of water in the system, limiting the participation of the hydroxyl group in the interaction. Alternatively, RS is limited by the fluorescence processes of aromatic residues of LTF and requires surface-enhanced procedures, e.g., by gold or silver nanoparticle sputtering [102]. Additionally, to accurately indicate the location of silver cation binding with LTF, Pomastowski et al. used molecular dynamics (MD) analysis. Moreover, by determining the locations of silver cation bonding with LTF, the reduction of silver ions to elementary silver via density functional theory (DFT) was indicated. On the other hand, X-ray photoelectron spectroscopy (XPS) was used to confirm the attachment of the LTF peptide (hLf1-11) to titanium surfaces by determining the chemical composition of the surface of the system [93]. Another important technique is fluorescence spectroscopy (FS) since it allowed one to determine binding affinity of whey proteins and metal ions by measurements of quenched fluorescence of the protein upon addition of metal. However, this method is suitable mostly for moderate and strong affinities and includes a number of details that are necessary to consider during the measurements [103]. Binding affinities of different metal ions and whey proteins were mostly determined by fluorescence spectroscopy [60,61,62,63,64,65,66]. The second method that was used for the determination of binding affinity of the interaction of metal ions and whey proteins was isothermal titration calorimetry (ITC). Obviously, it is not a spectroscopic method, but in contrast to FS, it was used only in two studies (Table 2). ITC is based on measurements of the heat changes during the interaction. One advantage of this method over other techniques is that it is possible to measure thermodynamic parameters of the interaction together with binding affinity, however for very high- or low-affinity complexes it is challenging [104,105].

#### 2.3.3. Microscopic Techniques 

Microscopic techniques can serve as indirect methods for study of consequences of interaction and description of the mechanism of metal ions binding to a protein leading to the formation of nanoparticles as it was carried out previously [47]. Kumari and Kondapi used FT-IR spectroscopy to confirm 5-FU entrapment in LTF nanoparticles in their studies on fluorouracil capture in lactoferrin nanoparticles to increase its effectiveness in the treatment of malignant melanoma. Transmission electron microscopy (TEM) and scanning electron microscopy (SEM) analyses were used for physicochemical characterization of LTF nanoparticles formed as a result of an interaction of LTF with silver ions. TEM analysis helped to determine the morphology and size of nanoparticles, while SEM helped to determine that the obtained particles have a spherical shape in the range of 90–110 nm. Additionally, studies based on dynamic light scattering showed that the hydrodynamic diameter of the LTF nanoparticles obtained was 150 ± 20 nm. These differences can be explained by the fact that dynamic light scattering (DLS) analysis takes into account the hydrodynamic size of the solvated protein, but electron microscopy approaches are used for the study of topology, porosity, and metal–organic core size [87]. However, Kumar et al. characterized the morphology of LTF nanoparticles using both SEM and atomic force microscopy (AFM), which showed that the nanoparticles had a spherical shape and a diameter of 50–60 nm [91]. In the study of Bollimpelli et al., DLS analysis of LTF nanoparticles loaded with curcumin was used to determine their size (100 nm), whereas SEM and AFM analyses indicated their sizes were 43–60 nm. In this case, different sizes were related to the surface charge of the particles and their interaction with the water shell [88].

#### 2.3.4. Complementarity of MALDI- and NALDI-TOF-MS for Metal–Protein Interactions Studies 

The matrix-assisted laser desorption ionization technique (MALDI) coupled to the time-of-flight mass spectrometry (TOF-MS) has become one of the common methods for protein characterization in addition to other MS-based approaches. The advantages of MALDI-TOF-MS include the simplicity of use, sensitivity, large mass range, and relative resistance to interferences from matrices [96]. Moreover, this technique is reasonable to use not only for protein characterization but also for studies of protein interactions because biological activity and biomolecular recognition are defined by non-covalent interactions. The interactions of various ligands with proteins are interesting to study because they fulfill many functions in nature as well as in the human body. For example, MALDI-TOF-MS analysis of β-lactoglobulin was used to study its influence on human immunity and promotion of cell proliferation [106], and the analysis of carbonic anhydrase IX [107] complexes with potential synthetic inhibitors allowed the identification of only strong binding inhibitors to proteins owing to the nature of MALDI ionization. Authors suggested that the stoichiometry of binding showed a possible second binding site, supporting the hypothesis of the induced-fit model of the interaction. In some cases, in studies of the origins of disease, the role of metal–protein complexes and metal speciation are of big interest, particularly for Alzheimer’s disease and its implications. Although MALDI-MS is not able to quantify metals in biological samples as well as in model systems for such studies, it can be used for quantitative determination of proteins and identification of post-translational modifications sites. Despite the challenges of MALDI-MS quantification associated with the reproducibility of results, approaches, such as using isobaric tags for relative and absolute quantization (iTRAQ), are widely used for differentiation of expressed proteins in comparative proteomics owing to large-scale, high throughput, and highly sensitive procedures of different MS-based approaches [108]. This technology applies a 4-plex set of amine reactive isobaric tags to derivatize peptides at the N-terminus and at the lysine side chains, thus labeling all peptides in a digest mixture [109]. 

Moreover, PTMs are crucial for protein biological activity, and their identification is an important analytical challenge. The MALDI-TOF-MS technique can be applied for this purpose by protein digestion to peptides and then identification of PTM sites. Although MALDI-TOF-MS is widely used for the analysis of digested proteins as well, analysis of LMW peptides can be complicated owing to the matrix interference and thus, signal suppression (<700 Da). Matrix-free approaches include many methods of analysis with the use of materials to replace the matrix. Nano-assisted laser desorption ionization (NALDI) is a promising technique. Nanomaterials assist the ionization that leads to signal enhancement owing to surface plasmon resonance. For example, Shenar et al. reported that a NALDI plate provided results with better sensitivity as compared to a DIOS chip, carbon powder, and porous silica for model peptides in the range of 519–2853 Da [110]. Interestingly, the NALDI approach can be promising for studies of metal–peptide interactions, which is also an important part of metal–protein interactions studies because they can indicate if some peptides can be specifically bound to a metal of interest.

Thus, the MALDI and NALDI techniques can be complementary to each other in terms of characterization of metal–protein interactions when coupled with other instrumental techniques in the following cases: (a) analysis and characterization of intact and digested proteins, analysis of metal–protein adducts by MALDI, (b) analysis of peptides digested from original proteins in the low molecular mass range, peptide adducts with metals and PTM sites located on the peptides with a low MW using NALDI in the mid to high range by MALDI, and (c) quantitative analysis of proteins by MALDI/NALDI. 

## 3. Implications of Whey Protein–Metal Interactions in Food and Nutraceuticals 

### 3.1. Changes in Bioactivity after Metal Interactions 

There are many publications that have reported biological claimed activities of whey proteins; however, the changes in these activities upon metal addition have not yet been deeply discussed. 

To the best of our knowledge, whey proteins exert antioxidant activity by forming glutathione (GSH) [111], which may be connected with anticancer activity. GSH can be formed due to high sulphur content of whey proteins provided by the presence of cysteine, which forms γ-glutamylcysteine, and this step is rate-limiting in GSH synthesis [112]. In addition, LTF can serve as an antioxidant agent by a binding iron in a form that prevents it to act as a Haber–Weiss catalyst [113]. However, complexes of whey proteins with metals demonstrate anticancer activity, serving as carriers for metal complexes, such as lanthanum (III) with β-LG [63] and cobalt and nickel complexes with HSA and β-LG [68].

Nevertheless, the effect of metals on anticancer activity of whey proteins can be shown by the example of LTF. It has been reported that iron-unsaturated apo-LTF inhibits the growth of cervical cancer (HeLa) cells after 48 h of treatment, whereas diferric-bLf was not effective [114]. Additionally, Gibbons et al. indicated that both iron-unsaturated apo-LTF and iron-saturated b-LTF (> 90% Fe^3+^ sat.) showed antitumorigenic properties; however, the apo-form showed a higher effect in inducing cytotoxicity in both cell lines (MDA-MB-231 and MCF-7) compared to Fe-bLf, which was more effective at inducing apoptosis in MCF-7 cell lines [115]. Studies conducted on mice by Kanwar et al. showed that chemotherapy eradicated EL-4 lymphomas in mice that received iron-saturated LTF for 6 weeks before chemotherapy but not in mice receiving lesser iron-saturated forms. It was concluded that bLf may be a potential natural adjuvant agent for supporting chemotherapy, but when saturated with iron, it can be more effective [116].

LTF possesses antibacterial activity itself along with having immunomodulatory functions [117]. However, in previous research by Pomastowski et al. [47], it was reported that the formed nanocomposite as a result of the interaction of silver ions with LTF and a spontaneous reduction of silver ions to nanoparticles exerted a strong antibacterial activity against *Pseudomonas aeruginosa, Staphylococcus aureus, Escherichia coli*, and *Enterococcus faecalis*. The growth of the drug-resistant *P. aeruginosa* strain was inhibited by more than 97%, which was comparable to the traditional cefotaxime antibiotic. Moreover, Komatsu et al. demonstrated that lactoferrin might be used to reduce the risk of aspiration pneumonia among elderly people for whom oral care remains difficult [118]. Additionally, the antiparasitic effect of LTF is different depending on the species. LTF interferes with iron acquisition by some parasites; however, LTF may act as a specific iron donor for some parasites that use LTF for their growth. More specifically, holo-LTF interacts with cell membrane receptors, and the cell starts to secrete iron reductase complex, which includes NADPH that requires the reductase, and it donates electrons, thus changing the membrane potential and leading to its disintegration [119]. 

An interesting study by Thawari et al. [120] on β-LG and apo-α-LA was reported on the effect of copper (II) ions on the formation of protein dimers and the catalytic activity of the formed nanobiomaterials, which were inorganic hybrid nanomaterials in its nature. Nanostructures in the form of nanoflowers and nanofibers combined with copper-LG and copper-LA with Cu as the inorganic part exhibited peroxidase-like catalytic activity. Important conclusion of the study was that β-LG produced a pH-dependent protein–protein dimer that formed at up to pH 12 and at a concentration of 1000 equiv. of copper (II), whereas in the case of α-LA, the formation of a dimer was not observed, which was explained by the non-availability of free Cys-SH compared to exposed Cys121 in β-LG. 

### 3.2. Consequences of Changes in Biological Activity in Food and Nutraceuticals 

Whey proteins as a component of milk and dairy products are a part of the everyday diet of many people. Together with the trend of uncontrolled consumption of supplements by the public, it creates a window for interactions between whey proteins and metals. Clearly, additional supplementation to food in the form of organic complexes that have metals can make up the deficiency of microelements, which has some advantages. Microelement deficiency is a serious problem that is common for developing countries, especially for infants, preschoolers, pregnant and lactating woman, and older adults [121], and in the long-term perspective it can cause chronic diseases and inflammation [122]. Since inorganic complexes with metals (e.g., chloride, etc.) have many side effects, they have been replaced by organic complexes with proteins and peptides that, in addition to minimizing the side effects, are more biologically available [123]. However, accidental interaction between WP and metals can be harmful because it can lead to unpredictable consequences. The formation of nanoparticles as a result of interactions, as it was discussed before, or the enhanced release of metals owing to the instability of the complex in gastric fluid can be toxic to humans. For example, zinc in the form of a dietary supplement is quite popular among modern consumers and has been studied for its chelating ability. It was found [61] that zinc was released from zinc–peptide complexes after simulated gastric digestion, thus confirming the suggestions that such complexes are not stable. However, two types of whey protein hydrolysate (WPH) were studied, the most negatively charged (1) and the least negatively charged (2), according to zeta potential measurements; and in a dispersion stability study, zinc release was found to be much higher for the second hydrolysate. The authors suggested that such differences in stability could be explained by the differences in the surface charge, particle size, and strength of the formed complexes. The molecular size of the peptides could also have an effect because smaller-sized peptides could reach the metal target more easily; however, both hydrolysates have a similar degree of hydrolysis and amount of peptide per mass. Moreover, zinc release increased with pancreatic digestion with a higher level of the second hydrolysate. Although similar studies were performed for casein, silver [124], and zinc-peptide complexes [125], the problem of the stability of metal–organic complexes as well as the unpredictable consequences of such interactions for the production of nutraceuticals has not been well-studied and deserves attention. In the study by Wang et al. [125], complexes of zinc and three peptides were almost unaffected under gastric conditions; however, the release of zinc ions was observed under pancreatic conditions; nevertheless, the zinc-Asn-Cys complex has potential for the improvement of zinc bioavailability. 

Another window for whey protein–metal interactions is the contact of milk products with stainless steel surfaces during storage, transport, and processing. Stainless steel alloys are of widespread use in the food industry owing to their high resistance to corrosion, good mechanical properties [126], and relative ease of cleaning. Although a few studies on metal release from stainless steel have been conducted, little attention has been paid to the effects of proteins on metal release, and data on these interactions during food storage and processing is scarce and not systemically investigated. Atapour et al. reported a study on metal release from stainless steel 316L under static and stirring conditions using a whey protein solution, simulated milk solution, and phosphate buffered saline solution for mechanistic comparison, and the results showed the release of iron, chromium, and nickel ion was much more significant for the whey protein solution, causing enhanced rates of protein aggregation and its precipitation from solution. Additionally, the authors remarked that the released concentrations of iron, chromium, and nickel did not exceed the limits estimated by European guidelines; nevertheless, the ratio of the solution volume of milk needs to be considered, and future toxicological assessments need to be performed. Alternative contact materials might not be better from a health perspective, but the effects of contact with milk needs to be investigated [127]. Then, the same research group studied the effect of the grade of corrosion on metal release using electrochemical methods, also confirming the importance of metal complexation by whey proteins on metal release [128].

One more important point is whey protein–metal interactions occurring in the process of purification of whey by magnetic metal nanoparticles that is a good alternative to membrane processes. It is crucial not only for the design of functional foods and its safety but also for utilization of whey because its utilization in the purified form is more safe compared to that of the raw form [129]. 

## 4. Conclusion

Whey proteins have a rich chemistry, thus providing interesting properties for many applications. Their natural affinity to metals has advantages as well as disadvantages in the fields of food chemistry and nutraceuticals. Advantages may include their potential applications as nutraceuticals, for the design of functional foods, whey purification, and utilization of whey wastes, while the drawbacks include their interactions with metals in food products, metals as drugs, and dietary supplements that may cause loss of the drug/supplement activity or toxic effects. This may occur with metals contact surfaces during processing, storage, and transport.

Moreover, the effect of metals on their biological activity deserves attention because this also creates opportunities for new applications and is an important factor to consider in all stages of food processing, food safety, quality, and utilization. Formation of complex biocolloidal systems as a result of metal–protein interactions may occur in different ways, such as via (i) metalloproteins, (ii) metallocomplexes, (iii) nanocomposites, and (iv) aggregated systems. This review is focused on a discussion of the biological activities, interactions with metals, and analytical techniques applied for their study. 

## Figures and Tables

**Figure 1 ijms-21-02156-f001:**
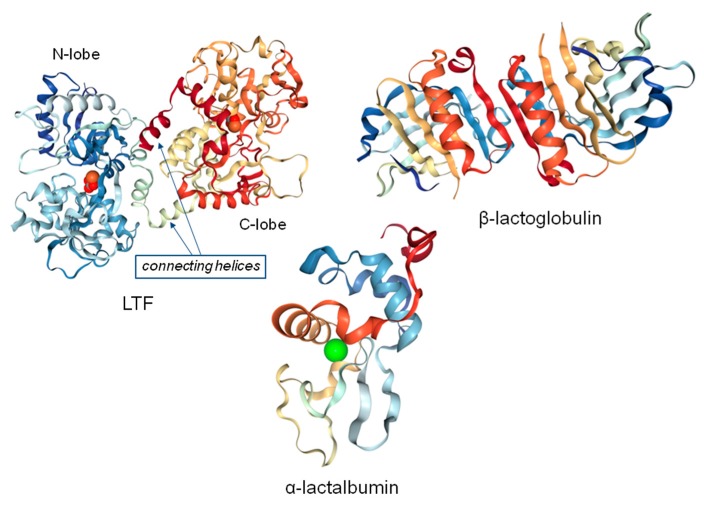
Structures of whey proteins at high resolution extracted from the protein data bank database; iron atoms are denoted in red color for LTF together with carbonate ions; calcium atom is denoted in green color for α-lactalbumin, PDB codes: LTF—1BIY; α-LA—1HFX; β-LG—6RYT.

**Figure 2 ijms-21-02156-f002:**
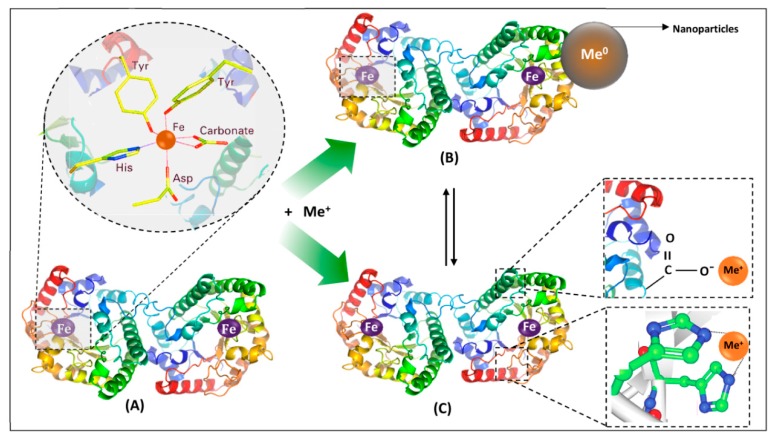
Consequences of metal–protein interactions: (**A**) metalloproteins, (**B**) nanoparticles, and (**C**) metallocomplexes; 2A—carbonate binding site of LTF as a metalloprotein; 2B—formed nanoparticles as a result of interaction of LTF with a metal ion; 2C—metallocomplexes formed by weak electrostatic and sandwich interactions; B and C can form a nanocomposite.

**Table 1 ijms-21-02156-t001:** Different whey protein molecular weights associated with post-translational modifications (PTMs).

Protein	Mol. Weight (kDa)	Theoretical mol. Weight (kDa)*	PTM	Method of Isolation/Purification	Identification	Ref.
β-LG	18	18.277	-	standard of β-LG (protein content > 90%)	SDS-PAGE MALDI-TOF-MS	[5]
	18.5					
β-LG	18.3	18.277	monomeric and the dimeric forms at pH 7.4 glycated β-lactoglobulin	β-LG was dissolved in 9.1 mM glucose in water, and the pH was adjusted to 7 with 50 mM phosphate buffer	MALDI-TOF-MS	[16]
	36.6					
β-LG	17.4	18.277	-	anion-exchange chromatography (DEAE-Sepharose)	SDS-PAGE	[17]
β-LG	19.9	18.277	proteins appeared as strings of spots, indicating their different isoforms with different charges as a result of PTMs occurring prior to secretion	precipitation via ammonium sulphate fractionation	2-DE	[18]
α-LA	16.2	16.247			MALDI-MS	
α-LA	14.1	16.247	small mass differences ruled out PTMs, such as phosphorylation and glycosylation	precipitation by ammonium sulphate	MALDI-TOF-MS	[19]
SA	67.7 (SA)	69.367	glycosylation of specific milk proteins was shown to vary during lactation; no potential N-glycosylation and O-linked glycans (SA), known N-linked glycoprotein (LTF)	0.5 mL of raw milk was centrifuged at 4 °C for 30 min, fat and cellular layers were removed; residual lipids were removed by addition of three volumes (1.5 mL) of 2:1 chloroform/methanol, agitation, retaining of supernatant; protein was precipitated from supernatant with ethanol overnight at 4 °C, followed by centrifugation; precipitate was re-suspended in 50 mM ammonium bicarbonate buffer (pH 7.5); glycans were separated by SDS-PAGE and extracted for MALDI-MS analysis	MALDI-MS	[20]
	79.8 (LTF)					
LTF	69.0 (SA)	78.056			LC–MS/MS	
	78.0 (LTF)					
LTF	80.002	78.056	-	milk was defatted by centrifugation, and the pH was then adjusted to 4.6 using hydrochloric acid; precipitated casein was removed by centrifugation	RP-LC–MS/MS	[21]

*values of theoretical molecular weight of the proteins from the Uniprot database (bovine, and for SA-human); Uniprot KB: α-LA—P00711; LTF—P24627; SA—P02768, the value for β-LG accounts for form B from publication of Eigel et al. [22].

**Table 2 ijms-21-02156-t002:** Interactions of metal ions with proteins and their characterization.

Metal/Conc.	Compound/Conc.	Interaction	Analytical method	Ref.
Zn^2+^		strong binding affinities:	ITC	[60]
	LTF	2.7 × 10^5^ M^−1^		
	BSA	2.3 × 10^5^ M^−1^		
	α-LA	1.5 × 10^5^ M^−1^		
	β-LG	1.5 × 10^5^ M^−1^		
Zn^2+^(6.23 mM)	α-LA (63.9 µM)	two sets of independent binding sites for zinc (II)	ITC	[34]
		two ions bind with the binding constant of 4.53 × 10^4^ M^−1^	fluorescence	
		four ions bind with the binding constant of 963 M^−1^	CD	
		electrostatic interactions	DSC	
Zn^2+^	whey-derived peptides	zinc chelation	FT-IR	[61]
		electrostatic interactions	zinc chelating capacity	
ZnO	WPI	DSC curves allowed to suggest; hydrogen bonding; O–Zn–O bonding; or electrostatic interactions; XRD and UV-Vis allowed to observe evidence for phase structure and crystal quality of ZnO nanoparticles; TEM—image of ZnO-WPI nanocomposite	XRD, TEM, DSC, UV-Vis	[62]
Ag^+^	LTF	two stages: (i) internal diffusion and sorption onto the external surface of lactoferrin globules; (ii) internal diffusion and binding to the lactoferrin structure; via electrostatic and hydrophobic interactions	MALDI-TOF/TOF-MS, ICP-MS, FT-IR, SERS, TEM, EDX, electrophoretic techniques	[47]
La (III)-Cys complex		hydrogen bonds, van der Waals interactions	NMR, UV-Vis, FT-IR, TG-DTA, FRET, CD	[63]
	BSA	K_BSA-La_ 0.11 × 10^4^ M^−1^;		
	β-LG	K_β__-LG-La_ 0.63 × 10^3^ M^−1^		
La (III)-Trp complex		hydrophobic interactions:	NMR, UV-Vis, FT-IR, TG-DTA	[64]
	HSA	K_b_ 0.138 × 10^4^ M^−1^ (303 K)		
La (III)-Phe complex		hydrogen bonds, hydrophobic interactions K_b_ 0.174 × 10^4^ M^−1^ (303 K)	NMR, UV-Vis, FT-IR	[65]
	HSA			
Pd (II) complex		hydrogen bonds, van der Waals interactions	NMR, UV-Vis, FT-IR	[66]
	HSA (1 × 10^5^ M)	K_b_ 0.5 × 10^4^ M^−1^;		
	β-LG (1 × 10^5^ M)	K_b_ 0.2 × 10^3^ M^−1^		
Pd (II) complexes (10^−4^ M)		hydrogen bonds, van der Waals interactions	NMR, UV-Vis, FT-IR, FRET	[67]
	HSA (2 mg/mL)	I complex: K_b_ 0.49 × 10^4^ M^−1^ (293 K);		
		II complex: K_b_ 0.79 × 10^4^ M^−1^ (293 K)		
Co (II)-Ni (II) complexes		hydrogen bonds, van der Waals interactions	UV-Vis, FT-IR, fluorescence	[68]
	HSA	K_b_ 3.16 × 10^6^ M^−1^ (303 K);		
	β-LG	K_b_ 0.54 × 10^5^ M^−1^ (303 K)		
Mn (II)-Co (II) complexes (5 × 10^−3^ M)		hydrogen bonds, hydrophobic interactions	UV-Vis, FT-IR, FRET	[69]
	HSA (5 × 10^−4^ M)	I: K_b_ 7.4 ± 0.04 × 10^4^ M^−1^ (303 K);		
		II: K_b_ 6.08 ± 0.09 × 10^3^ M^−1^ (303 K)		
	β-LG (5 × 10^−4^ M)	I: K_b_ 7.13 ± 0.03 × 10^4^ M^−1^ (303 K);		
		II: K_b_ 2.62 ± 0.05 × 10^3^ M^−1^ (303 K)		

**Table 3 ijms-21-02156-t003:** Analytical techniques used for the separation and identification of whey proteins.

Proteins	Matrix	Isolation	Separation	Identification	Ref.
α-LA	cheese	cheese extracts were desalted and preconcentrated using microcon membranes	CE with fused silica capillaries	DAD	[73]
β-LG A					
β-LG B					
β-LG	cow, goat, and ewe cheeses, incl. those of a single animal origin, binary ternary mixtures	desalted, preconcentrated samples were obtained with microcon membranes	CE with fused silica uncoated capillaries	DAD	[74]
α-LA					
α-LA	raw milk	mixture of standards of purified proteins, separation was achieved by adding SEP and TPS buffers to milk	SDS-PAGE; Microfluidic chip electrophoresis	Fluorescence	[75]
β-LG					
α-LA	fresh skim milk	mixed protein standards were prepared by combining each of the individual protein solutions (1 mL)	SDS-PAGE; Microfluidic chip electrophoresis	Fluorescence	[76]
β-LG					
caseins					
β-LG	milk	diluting 200 µL of ultracentrifuged whey with 400 µL of HPLC-grade water	LC, Jupiter C4 column; Microchip electrophoresis	UV, MS; Fluorescence	[77]
α-LA					
SA					
LTF	milk	samples were centrifuged to remove fat; skim milk was loaded onto lactoferrin immunoaffinity column	LC, Symmetry C4 Column	PDA	[78]
β-LG					
α-LA					
β-LG	buffalo mozzarella	mixtures of cow’s milk, water buffalo’s milk, mixtures of brine from cow’s milk mozzarella, brine from buffalo mozzarella were prepared in diff. vol. ratios for calibration purposes	LC, Supelco Discovery Bio Wide Pore C8 column	MS	[79]
α-LA	WPC	standard pure proteins to determine ret. times; equilibration buffer Tris-HCl at 20 mM; elution buffer Tris-HCl at 20 mM with 1 M NaCl were used for separation	Mono Q5/50 GL anion-exchange column, FPLC	UV-Vis; SDS-PAGE	[80]
β-LG					
BSA					
α-LA	mozzarella cheese whey	different equilibration and elution buffers were prepared	Chromatographic column; packed with SP Sepharose Big; Beads cation exchanger, HPLC	UV-Vis; SDS-PAGE	[81]
β-LG					
BSA					

**Table 4 ijms-21-02156-t004:** Instrumental techniques used to study the metal–protein interactions and their potential applications.

Compounds	Form	Application	Analytical methods	Ref.
LTF	metalloprotein	regulation of inflammation and oxidative stress in vertebrates	AFM	[83]
α-LA	metalloprotein	nutrition of infants in a long breastfeeding stage	Native-PAGE; SEC-ICP-MS; MALDI-TOF/TOF-MS	[84]
LF				
serum albumin				
LTF	nanoparticles	gene delivery carrier with targeting abilities	TEM	[85]
WPI	nanoparticles	production of antimicrobial cotton fabrics	UV-Vis; TEM; SEM	[86]
LTF	nanoparticles	increased therapeutic efficacy of treatment of malignant melanoma	TEM; SEM; DLS; FT-IR	[87]
LTF	metallocomposites; nanoparticles	in medicine and food industry as an antimicrobial agent	MALDI-TOF/TOF-MS; ICP-MS; FT-IR; SERS; TEM; I-DE; zeta potential measurements	[47]
LTF	nanoparticles	drug delivery strategy against the neurotoxicity in dopaminergic neurons	FE-SEM; AFM; DLS	[88]
LTF	metalloprotein	a therapeutic lead for controlling neutrophil extracellular traps (NETs) release in autoimmune and inflammatory diseases	TEM; SEM; fluorescence microscopy; agarose gel electrophoresis	[89]
IgA	metalloprotein	health and nutrition of breastfed newborns	ESI-MS/MS; FAAS	[90]
LTF	nanoparticles	target specific drug delivery, encapsulation of the drug	FE-SEM; AFM; FT-IR	[91]
LTF	nanoparticles	drug delivery for effective targeting therapy of brain glioma	Particle electrophoresis	[92]
BSA				
LTF	metalloprotein	antimicrobial biomaterials for dental applications	HPLC; SEM; XPS	[93]
LTF	metalloprotein	inhibition of the attachment of free HIV-1 to epithelial cells	ELISA; flow cytometry	[94]
LTF	Metalloprotein; metallocomplex	immobilized DNA effective for LTF purification	HPCEC; HPIMAC; HPLC; HPSEC; SDS-PAGE	[95]

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
