# Peer review of "Interactions of Whey Proteins with Metal Ions"

_ijms, 2020, doi:10.3390/ijms21062156_

Round 1

Reviewer 1 Report

This review described in a first part the technics and methods used to purify and identify whey proteins (mainly lactoferrin, b-lactoglobulin and a-lactalbumin), then the analytical methods to study their interaction with metals and finally the roles of protein-metal interactions in their biological function and in food and nutraceuticals. This work is clear and interesting, but maybe it missing some data. So, I have some comments and suggestions before the publication in International Journal of molecular Sciences.

Minor remarks:

  • line 15, “which have” not has
  • line 44 and 47, Use the same spelling for disulphide and disulfide. Same remark for oxidized or oxidized (lane 211)
  • The abbreviation PTM should appear lane 49. 5-FU is missing in the abbreviations list
  • I suggest that the abbreviations be listed alphabetically, because of their large number.
  • In table 2, line 3, WPH should be replaced by WPI, right?
  • Buffer TPS and SEP should be described
  • Table 4, row IgA, remove the end parenthesis in column 4
  • A reference is missing at the end of line 365

Major remarks:

  • First paragraph, page 5, the authors refer to the article of Mercandate et al [16]. I don’t understand the mention of “calcium chemistry” (line 69). They also wrote that “the equilibrium constant constant increased with an increase in pH”, but in this article, KD increased with the increase of the difference of pH and pI, which in not the same.
  • Page 6, line 117. It is surprising that “the affinity to iron … is enhanced when the pH becomes slightly acidic”. No mention of that in the reference proposed [40]. In fact, lactoferrin releases its iron at pH lower than 4. (Med Hypotheses.2012 Aug;79(2):219-21. doi: 10.1016/j.mehy.2012.04.044.)
  • Page 6, last paragraph, the authors described the difference between metalloprotein, metallocomplexe, and nanocomposites. Is their definitions? Or could they give a reference about it?
  • About the metals in interaction with proteins (page 8), it may be interesting to give some values: the concentration of some metals in milk, and the affinity constant of metalloproteins. Because for example, when the authors affirmed that BSA has strong Zn binding with an affinity constant of ~104 M-1, it is very low compared to iron with lactoferrin (~1023 M-1) (see ref 52). A reference is missing at the end of line 183 about metallocomplexes as cancer drugs. In the table 2, the oxidation number should be mentioned for all the metal.
  • In the paragraph “Nature of metal-protein interaction”: last paragraph, I suggest to change “amino and carbonyl groups of proteins” by “of polypeptide chains”. Don’t forget the nitrogen of Lys and Arg side chains which are also involved in metal interaction.
  • In the “analytical techniques for studies of interactions of whey proteins with metal ions” part, I think two main analytical techniques are missing: ITC and fluorescence. See for example J Dairy Res 2012;79(2):209-15 and J Biomol Struct Dyn; 2019;37(8):2072-85. Microscopic techniques are not appropriate for metal-interaction studies. In this paragraph, no example of metal-interaction is described.
  • In my opinion, the paragraph 2.4. should be in the paragraph 3.
  • About the anti-oxydant activity of whey proteins (lane 377), it seems that lactoferrin has a role in oxidative stress, because it avoids the Fenton and HaberWeiss reactions by sequestering iron (Adv Exp Med Biol. 1994;357:143-56.)

Author Response

We are very grateful to your critical comments and thoughtful suggestions. Based on these comments and suggestions, we have made a careful revision of the original manuscript. A revised manuscript has been submitted, in which the modified sections are highlighted in red. Thank the Editor and Reviewers again, who made great contributions to improve our paper.

Point 1:  line 15, “which have” not has

Response 1: This mistake has been corrected.

Point 2: line 44 and 47, Use the same spelling for disulphide and disulfide. Same remark for oxidized or oxidized (lane 211)

Response 2: This spelling mistake has been corrected in lines 44 and 47 as well as in the rest of the text. 'Oxidized' was checked throughout the text and was found only in line 211, which was corrected by native English speaker during the language check and the option 'oxidised' was selected for the text.

Point 3: The abbreviation PTM should appear lane 49. 5-FU is missing in the abbreviations list.

Response 3: The abbreviation PTM was introduced in the text in line 49. 5-FU was added to abbreviations list.

Point 4: I suggest that the abbreviations be listed alphabetically, because of their large number.

Response 4: Thank you for Your suggestion, it has been followed and abbreviations were organized alphabetically.

Point 5: In table 2, line 3, WPH should be replaced by WPI, right?

Response 5: In table 2, line 3 the reference 60 (Udechukwu et al., Food Chem 240, 1227-1232, 2018) was addressed and authors studied zinc chelating capacities with whey protein hydrolysates and their complexes stabilities.

Point 6: Buffer TPS and SEP should be described

Response 6: Buffers were added to abbreviation list not to overload the table.

Point 7: Table 4, row IgA, remove the end parenthesis in column 4

Response 7: Thank You for your comment, the end parenthesis was deleted.

Point 8: A reference is missing at the end of line 365

Response 8: Mentioned sentence is a critical remark of the authors on perspectives of application of nano-assisted laser desorption ionization technique for analysis of metal-protein interactions. By this sentence authors wanted to say that it could be possible to see which peptide bind to metal and suggest stoichiometry by changes in molecular mass of the peptides.

Point 9: First paragraph, page 5, the authors refer to the article of Mercandate et al [16]. I don’t understand the mention of “calcium chemistry” (line 69). They also wrote that “the equilibrium constant constant increased with an increase in pH”, but in this article, KD increased with the increase of the difference of pH and pI, which in not the same.

Response 9: Thank you for your comment. Indeed, calcium chemistry is not related to dimers and oligomers of β-LG formation, thus this claim was deleted. 'With the increase of pH' was corrected to 'with the increase in '|pH-pI|'. Authors apologize for this inaccurate claim. 

Point 10: Page 6, line 117. It is surprising that “the affinity to iron … is enhanced when the pH becomes slightly acidic”. No mention of that in the reference proposed [40]. In fact, lactoferrin releases its iron at pH lower than 4. (Med Hypotheses.2012 Aug;79(2):219-21. doi: 10.1016/j.mehy.2012.04.044.)

Response 10: Thank you very much for this comment. Indeed, the mistake with reference 40 has been made. Authors revised the part of the text related to the affinity of LTF to iron as a function of pH with addition of the proposed reference.

Point 11: Page 6, last paragraph, the authors described the difference between metalloprotein, metallocomplexe, and nanocomposites. Is their definitions? Or could they give a reference about it?

Response 11: The definition of metalloprotein mentioned in the text is common, but the remarks about metallocomplexes and nanocomposites, formed as a result of interaction of whey proteins and metal ions, were made according to previous works of authors in the field of sorption of metal ions onto protein and synthesis of nanocomposites of silver and protein complex with zinc. Paragraph was revised and corresponding references were added.

Point 12: About the metals in interaction with proteins (page 8), it may be interesting to give some values: the concentration of some metals in milk, and the affinity constant of metalloproteins. Because for example, when the authors affirmed that BSA has strong Zn binding with an affinity constant of ~104 M-1, it is very low compared to iron with lactoferrin (~1023 M-1) (see ref 52). A reference is missing at the end of line 183 about metallocomplexes as cancer drugs. In the table 2, the oxidation number should be mentioned for all the metal.

Response 12: The values of affinity constants (where available) and concentrations of both metal ion and protein were added to the table for readers comparison between the interactions.

Point 13: In the paragraph “Nature of metal-protein interaction”: last paragraph, I suggest to change “amino and carbonyl groups of proteins” by “of polypeptide chains”. Don’t forget the nitrogen of Lys and Arg side chains which are also involved in metal interaction.

Response 13: Your recommendation has been followed and 'amino and carbonyl groups of proteins' was replaced to 'polypeptide chains' with corresponding changes in the sentence.

Point 14: In the “analytical techniques for studies of interactions of whey proteins with metal ions” part, I think two main analytical techniques are missing: ITC and fluorescence. See for example J Dairy Res 2012;79(2):209-15 and J Biomol Struct Dyn; 2019;37(8):2072-85. Microscopic techniques are not appropriate for metal-interaction studies. In this paragraph, no example of metal-interaction is described.

Response 14: ITC and fluorescence were added to the section 'analytical techniques for studies of interactions of whey proteins with metal ions'. Microscopic techniques were described in this section since they are indirect methods to describe the mechanism of complex formation and their application allows to observe not the interaction itself, but its consequences such as formed nanoparticles or nanocomposite that is, in opinion of authors, important for assessment of these interactions as well as its implications in everyday life. However, the paragraph, dedicated to microscopic techniques, was revised to add more clarity for this point. You may also want to be redirected to our previously published papers about silver-LTF nanocomplexes (Pomastowski et al., 2016; Silver-Lactoferrin Nanocomplexes as a Potent Antimicrobial Agent. Journal of the American Chemical Society, 138(25), 7899–7909. https://doi.org/10.1021/jacs.6b02699).

Point 15: In my opinion, the paragraph 2.4. should be in the paragraph 3.

Response 15: Thank you for this comment. Paragraph 2.4. was moved to section 3.  

Point 16: About the anti-oxydant activity of whey proteins (lane 377), it seems that lactoferrin has a role in oxidative stress, because it avoids the Fenton and HaberWeiss reactions by sequestering iron (Adv Exp Med Biol. 1994; 357:143-56.)

Response 16: Thank you very much for your interesting comment. This point of the role of LTF in oxidative stress was added to the text.

Reviewer 2 Report

In this work, the authors aim to provide a fairly thorough review of what is known about the interaction of whey proteins with metal cations, with an emphasis on biologically relevant d-block metals. The authors discuss fundamental aspects of metal-protein interaction, the analytical tools used to study these interactions, and effects of metal binding, including implications for food and nutraceuticals. Two major omissions in the current version of the manuscript are discussions of (1) the effect of pH on protein structure and metal binding, and (2) the use of electrospray ionisation – mass spectrometry as an alternative to MALDI-MS. I have provided a more detailed list of comments below, including some regarding language and presentation. I believe this review article could eventually become a good addition to the field and worthy of publication, once these points have been addressed.

General comment: The English in this manuscript is understandable, but the text contains numerous idiosyncrasies and outright spelling and grammatical errors. I have pointed out a few of these below, but I highly recommend having a native or highly proficient English speaker proofread and edit the manuscript to bring it up to a publishable standard.

In the Introduction, each protein considered in this review is discussed in turn. This is a good approach, but it would be helpful if subheadings (1.1. beta-lactoglobulin; 1.2. alpha-lactalbumin; etc.) were explicitly introduced to make this structure clear to the reader

Also in the Introduction, the 3D structure of the major whey proteins is described. The structure of lactoferrin is used as the basis for the cartoon in Figure 1 (although the protein isn’t explicitly identified in the text or caption, but the structures of the other proteins are not shown at any point in the manuscript. It would be helpful to add a figure with the high-resolution structures of all proteins discussed in the manuscript, rather than just a description of the structure in the text.

Page 2, line 33; and page 25, abbreviation list: ‘lactoproxidase’ – I believe ‘lactoperoxidase’ is more standard.

Page 2, line 46: ‘d-electron metal’, and also ‘d-metal’ in other parts of the manuscript. Please replace all instances of these phrases with ‘d-block metal’, which is the standard term.

Page 2, line 47: ‘Thanks to’ is unscientific language and should be replaced with something more neutral, for example ‘due to’.

Page 2, lines 48-52: Several issues here. First, delete ‘which are examples […] one or more amino acids’ – it’s not necessary to explain the concept of a PTM. Second, I agree that the molecular weight of a protein depends on PTMs, but it certainly does not depend on the analytical method used. Different methods will potentially give answers with different degrees of precision and accuracy, but the fact remains that a given proteoform has one well-defined molecular weight. If a mix of proteoforms with different mass values is present, then separation/purification can skew this distribution and in this way the masses observed in the purified fraction will depend on the purification protocol, but this is merely a methodological artefact. The current way this is phrased in the manuscript is very confusing. The same goes for page 5, line 105-106.

Page 2, line 55: ‘irreversible modifications of monomers between 60 and 70 °C’. Could you provide specific examples of such modifications?

Table 1: The number of significant digits used in the mass values in the second column is not justifiable. For instance, the very first value (18.000 kDa) is presumably based on a statement in reference [5] that the protein has a mass of 18 kDa. However, the only value reported to a 1-Da accuracy in that paper is the second one (18.462 kDa), so the 18.000 kDa value is nonsense. It would be better to report values with an accuracy of 0.1 kDa consistently. Even then, I am unconvinced that the SDS-PAGE results are reliable (even if some authors, like those of reference [32], are unaware of this and have reported GE results to an even greater accuracy), but this is not the responsibility of the authors, who can only report what was in the original papers. I would, however, recommend adding the theoretical mass based on the sequence (easy to find in Uniprot or other databases) so the reader can compare this to the measured values.

Page 5, line 91: ‘the zinc(II)-binding site’ [of alpha-LA] is mentioned as a binding site for Mg(II) that is preferred over the Ca(II)-binding site, but an actual description of this site seems to be missing. Then, in line 97, it is implied that Zn(II) and Mg(II) actually do bind to the Ca(II)-binding site, which seems to contradict the earlier statement the way it is currently phrased. Finally, on line 101-103 it is stated that binding of Ca(II) increases the protein’s resistance to thermal treatment. This sentence should probably be moved up to around line 90, where the structural effect of removal of Ca(II) is discussed.

Page 6, line 113: the synergistic binding of carbonate that accompanies binding of iron to lactoferrin is mentioned; however, it would be good to briefly describe the way iron is coordinated in this complex (including the sites occupied by carbonate or bicarbonate anions).

Page 6, lines 123-138: There does not seem to be a single reference in this section, despite quite a few concepts being introduced in these two paragraphs. Please find and add some appropriate references.

Page 6, line 132-133: Please add a proper definition of the word ‘metallocomplex’, similar to that provided for ‘metalloprotein’ (‘nanoparticle’ is fairly self-evident and doesn't really need a definition).

Page 7, line 157: ‘fouling of the protein’. I assume that ‘fouling [of something] by the protein’ is meant?

Page 8, line 174-175: ‘coordinate in contrast to electron-rich proteins…’ This is a bit ambiguous, so I suggest replacing with ‘coordinate to electron-rich moieties in proteins…’

Page 8, line 177: ‘Moreover […] molybdenum(II)’. This whole sentence makes no sense and needs to be rewritten.

Page 8, line 182: ‘cancer (complexes of palladium and ruthenium)’. By far the most common metal in chemotherapy is platinum (see for example Butler & Sadler, Curr Opin Chem Biol, 2013, 175-188) so why is this metal not mentioned?

Section 2.1: Overall the text here is good (although there is some redundancy in the discussion on lines 207-212, which can probably be condensed to one sentence), but there does not seem to be much discussion of the effect of pH. This can have a major effect on metal binding, both directly by (de)protonation of important moieties, as well as indirectly by affecting whether the protein is folded in the correct way so that these moieties are arranged to form a binding site. Given the importance of pH, I would expect this factor to be discussed here, particularly when discussing which amino acids are likely to be important (depending on pH, these may or may not have free electron pairs available).

Table 2: The abbreviation ‘WPH’ seems to lack a definition in the manuscript.

Page 11, line 216: ‘fluorescence of beta-LG and BSA’ – please specify that this was the fluorescence of the Trp fluorophore

Table 3: Something seems to have gone wrong with the layout of the ‘matrix’ column, and everything from ‘buffalo mozzarella’ onward needs to be shifted down by one line

Page 14, line 245: ‘has been reported’ – please provide a reference

Section 2.3.1: The authors do a good job describing MALDI- and ICP-MS, but the gold standard for the determination of the accurate mass, sequence, and PTMs of a protein is electrospray ionisation – mass spectrometry. Native ESI-MS has the benefit of allowing the observation of even relatively fragile complexes, unlike MALDI, where only extremely strong noncovalent binding can be observed. For example, Allen (Anal Chem, 2013, 12055-12061; and Analyst, 2016, 884-891) describes native MS of beta-LG, LTF, and holo-transferrin, which seems highly relevant to this review. These techniques, and possibly also ion mobility, need to be discussed.

Page 18, line 306: ‘functional density theory’ - this should be ‘density functional theory’

Page 18, line 313: ‘electron transmission microscopy’ - this should be ‘transmission electron microscopy’

Section 2.3.4: Again, while important progress has been made in recent years to allow the study of noncovalent complexes via desorption ionisation, native ESI-MS remains the gold standard. In this regard, the excellent review by Leney & Heck (J Am Soc Mass Spectrom, 2017, 5-13) should be pointed out to the reader. Where the effect of metals on Alzheimer’s disease is mentioned (line 343) it would be worth citing some recent papers using MS to study the interaction of metals with beta-amyloid, for example Lermyte, J Am Soc Mass Spectrom, 2019, 2123-2134 and/or Lermyte, Cells, 2019, 1231

Page 19, lines 353 onward: While MALDI is indeed useful for proteomics, again, LC-MSMS has been the standard method for a long time and should also be discussed (see e.g. Aebersold & Mann, Nature, 2003, 198-207; Aebersold & Mann, Nature, 2016, 347-355).

Page 20, line 377: ‘whey proteins exert anti-oxidant activity by forming glutathione’ – this seems very unlikely. Please add a reference and clarify what is meant here.

Author Response

We are very grateful to Your critical comments and thoughtful suggestions. Based on these comments and suggestions, we have made a careful revision of the original manuscript. A revised manuscript has been submitted, in which the modified sections are highlighted in red. Thank the Editor and Reviewers again, who made great contributions to improve our paper.

Point 1: The English in this manuscript is understandable, but the text contains numerous idiosyncrasies and outright spelling and grammatical errors. I have pointed out a few of these below, but I highly recommend having a native or highly proficient English speaker proofread and edit the manuscript to bring it up to a publishable standard.

Response 1: Thank You so much for Your comment. All Your recommendations regarding spelling and grammatical errors have been followed. The text of the manuscript has been corrected by english native speaker.

Point 2: In the Introduction, each protein considered in this review is discussed in turn. This is a good approach, but it would be helpful if subheadings (1.1. beta-lactoglobulin; 1.2. alpha-lactalbumin; etc.) were explicitly introduced to make this structure clear to the reader

Response 2: Thank You for Your comment. The text in introduction was separated to subheadings for each protein to make the structure more clear to the reader.

Point 3: Also in the Introduction, the 3D structure of the major whey proteins is described. The structure of lactoferrin is used as the basis for the cartoon in Figure 1 (although the protein isn’t explicitly identified in the text or caption, but the structures of the other proteins are not shown at any point in the manuscript. It would be helpful to add a figure with the high-resolution structures of all proteins discussed in the manuscript, rather than just a description of the structure in the text.

Response 3: Authors understand the point, structures of proteins were added as a new figure. Caption of Figure 1 (Figure 2 in revised version) was updated.

Point 4: Page 2, line 33; and page 25, abbreviation list: ‘lactoproxidase’ – I believe ‘lactoperoxidase’ is more standard.

Response 4: Thank You for Your comment. The recommendation has been followed.

Point 5: Page 2, line 46: ‘d-electron metal’, and also ‘d-metal’ in other parts of the manuscript. Please replace all instances of these phrases with ‘d-block metal’, which is the standard term.

Response 5: Thank You for Your comment. The recommendation has been followed throughout all text.

Point 6: Page 2, line 47: ‘Thanks to’ is unscientific language and should be replaced with something more neutral, for example ‘due to’.

Response 6: Thank You for this comment. 'Thanks to' was corrected to 'due to'.

Point 7: Page 2, lines 48-52: Several issues here. First, delete ‘which are examples […] one or more amino acids’ – it’s not necessary to explain the concept of a PTM. Second, I agree that the molecular weight of a protein depends on PTMs, but it certainly does not depend on the analytical method used. Different methods will potentially give answers with different degrees of precision and accuracy, but the fact remains that a given proteoform has one well-defined molecular weight. If a mix of proteoforms with different mass values is present, then separation/purification can skew this distribution and in this way the masses observed in the purified fraction will depend on the purification protocol, but this is merely a methodological artefact. The current way this is phrased in the manuscript is very confusing. The same goes for page 5, line 105-106.

Response 7: Thank You for so constructive comment. Firstly, the part with explanation of the concept of PTM was deleted. Secondly, the claim 'molecular weight of the protein depends on PTMs' was revised, according to Your precise remark.

Point 8: Page 2, line 55: ‘irreversible modifications of monomers between 60 and 70 °C’. Could You provide specific examples of such modifications?

Response 8: Mentioned modifications have been clarified and the paragraph was revised considering another important factor of pH on thermal behaviour of the protein.  

Point 9: Table 1: The number of significant digits used in the mass values in the second column is not justifiable. For instance, the very first value (18.000 kDa) is presumably based on a statement in reference [5] that the protein has a mass of 18 kDa. However, the only value reported to a 1-Da accuracy in that paper is the second one (18.462 kDa), so the 18.000 kDa value is nonsense. It would be better to report values with an accuracy of 0.1 kDa consistently. Even then, I am unconvinced that the SDS-PAGE results are reliable (even if some authors, like those of reference [32], are unaware of this and have reported GE results to an even greater accuracy), but this is not the responsibility of the authors, who can only report what was in the original papers. I would, however, recommend adding the theoretical mass based on the sequence (easy to find in Uniprot or other databases) so the reader can compare this to the measured values.

Response 9: The values of molecular weight of proteins have been corrected in accordance with precision of the used method. All values were corrected to accuracy 0.1 kDa, and only one paper reported the value to 0.1 Da accuracy and the value, that was reported to account for 80.002 kDa, was left without changes.

Point 10: Page 5, line 91: ‘the zinc(II)-binding site’ [of alpha-LA] is mentioned as a binding site for Mg(II) that is preferred over the Ca(II)-binding site, but an actual description of this site seems to be missing. Then, in line 97, it is implied that Zn(II) and Mg(II) actually do bind to the Ca(II)-binding site, which seems to contradict the earlier statement the way it is currently phrased. Finally, on line 101-103 it is stated that binding of Ca(II) increases the protein’s resistance to thermal treatment. This sentence should probably be moved up to around line 90, where the structural effect of removal of Ca(II) is discussed.

Response 10: Thank You for Your comment. Calcium-binding site was introduced to the text in lines 87-89 (in the revised version of the manuscript). Indeed, the sentences below seems to be contradictory and second one was re-phrased. The sentence about structural effect upon removal of Ca (II) was moved to lines 102-104 (in a new version of the manuscript).

Point 11: Page 6, line 113: the synergistic binding of carbonate that accompanies binding of iron to lactoferrin is mentioned; however, it would be good to briefly describe the way iron is coordinated in this complex (including the sites occupied by carbonate or bicarbonate anions).

Response 11: Thank You for this comment. The way of iron coordination to LTF was briefly discussed.

Point 12: Page 6, lines 123-138: There does not seem to be a single reference in this section, despite quite a few concepts being introduced in these two paragraphs. Please find and add some appropriate references.

Response 12: The first two paragraphs does not contain references, since there is a critical summary of work has been made by authors, but all the references were given below connected to specific claims, regarding particular interactions.

Point 13: Page 6, line 132-133: Please add a proper definition of the word ‘metallocomplex’, similar to that provided for ‘metalloprotein’ (‘nanoparticle’ is fairly self-evident and doesn't really need a definition).

Response 13: Definition of 'metallocomplex' is proposed by authors based on previous experience in the field of interactions of proteins with metal ions. Reference to these works were given in the text and the paragraph was considerably revised. Caption for Figure 1 was also updated.

Point 14: Page 7, line 157: ‘fouling of the protein’. I assume that ‘fouling [of something] by the protein’ is meant?

Response 14: Thank You for Your comment. 'Fouling of protein' was corrected to 'fouling of filtration membrane by protein'.

Point 15: Page 8, line 174-175: ‘coordinate in contrast to electron-rich proteins…’ This is a bit ambiguous, so I suggest replacing with ‘coordinate to electron-rich moieties in proteins…’

Response 15: Your recommendation has been followed and the mentioned claim was corrected to 'to coordinate to electron-rich moieties in proteins'. Authors wanted to write that d-block metals are electron-deficient making them able for coordination bonds in comparison with proteins rich in electrons. However, this claim indeed looks ambiguous and has been corrected.

Point 16: Page 8, line 177: ‘Moreover […] molybdenum(II)’. This whole sentence makes no sense and needs to be rewritten.

Response 16: Thank You for constructive comment. The whole sentence was re-written.

Point 17: Page 8, line 182: ‘cancer (complexes of palladium and ruthenium)’. By far the most common metal in chemotherapy is platinum (see for example Butler & Sadler, Curr Opin Chem Biol, 2013, 175-188) so why is this metal not mentioned?

Response 17: Thank You for this comment. Authors mentioned palladium and ruthenium since their interactions with whey proteins were discussed below to give the readers more convenient order of information in the current review. However, application of platinum as anticancer agent was also mentioned in the revised version of the manuscript.

Point 18: Section 2.1: Overall the text here is good (although there is some redundancy in the discussion on lines 207-212, which can probably be condensed to one sentence), but there does not seem to be much discussion of the effect of pH. This can have a major effect on metal binding, both directly by (de)protonation of important moieties, as well as indirectly by affecting whether the protein is folded in the correct way so that these moieties are arranged to form a binding site. Given the importance of pH, I would expect this factor to be discussed here, particularly when discussing which amino acids are likely to be important (depending on pH, these may or may not have free electron pairs available).

Response 18: Thank You for this important comment. Discussion of the effect of pH on interactions has been added to the text.

Point 19: Table 2: The abbreviation ‘WPH’ seems to lack a definition in the manuscript.

Response 19: The abbreviation 'WPH' was added to abbreviations list and also was defined upon its first appearing in the text (line 432, page 21).

Point 20: Page 11, line 216: ‘fluorescence of beta-LG and BSA’ – please specify that this was the fluorescence of the Trp fluorophore

Response 20: Thank You for Your comment. Fluorescence of Trp fluorophore was specified in the mentioned sentence.

Point 21: Table 3: Something seems to have gone wrong with the layout of the ‘matrix’ column, and everything from ‘buffalo mozzarella’ onward needs to be shifted down by one line

Response 21: Thank You for this point. The mistakes with layout in Table 3 have been corrected.

Point 22: Page 14, line 245: ‘has been reported’ – please provide a reference

Response 22: Thank You for comment. The reference has been provided.

Point 23: Section 2.3.1: The authors do a good job describing MALDI- and ICP-MS, but the gold standard for the determination of the accurate mass, sequence, and PTMs of a protein is electrospray ionisation – mass spectrometry. Native ESI-MS has the benefit of allowing the observation of even relatively fragile complexes, unlike MALDI, where only extremely strong noncovalent binding can be observed. For example, Allen (Anal Chem, 2013, 12055-12061; and Analyst, 2016, 884-891) describes native MS of beta-LG, LTF, and holo-transferrin, which seems highly relevant to this review. These techniques, and possibly also ion mobility, need to be discussed.

Response 23: Thank You for this comment. Authors do understand that ESI technique is a gold standard in determination of accurate mass, sequence and PTMs of the protein. Discussion of ESI-MS in this sense has been added to the text.  

Point 24: Page 18, line 306: ‘functional density theory’ - this should be ‘density functional theory’

Response 24: This mistake has been corrected.

Point 25: Page 18, line 313: ‘electron transmission microscopy’ - this should be ‘transmission electron microscopy’

Response 25: This mistake has been corrected.

Point 26: Section 2.3.4: Again, while important progress has been made in recent years to allow the study of noncovalent complexes via desorption ionisation, native ESI-MS remains the gold standard. In this regard, the excellent review by Leney & Heck (J Am Soc Mass Spectrom, 2017, 5-13) should be pointed out to the reader. Where the effect of metals on Alzheimer’s disease is mentioned (line 343) it would be worth citing some recent papers using MS to study the interaction of metals with beta-amyloid, for example Lermyte, J Am Soc Mass Spectrom, 2019, 2123-2134 and/or Lermyte, Cells, 2019, 1231

Response 26: Thank You for this comment. Mentioned references were used for discussion.

Point 27: Page 19, lines 353 onward: While MALDI is indeed useful for proteomics, again, LC-MSMS has been the standard method for a long time and should also be discussed (see e.g. Aebersold & Mann, Nature, 2003, 198-207; Aebersold & Mann, Nature, 2016, 347-355).

Response 27: Thank You for comment. Mentioned paragraph has been revised.

Point 28: Page 20, line 377: ‘whey proteins exert anti-oxidant activity by forming glutathione’ – this seems very unlikely. Please add a reference and clarify what is meant here.

Response 28: Thank You for this comment. The reference was added.

Round 2

Reviewer 1 Report

The manuscript is now ready for publication.

Author Response

Response to Reviewer 1 Comments

Ref. No.:  ijms-732358

Interactions of whey proteins with metal ions

Agnieszka Rodzik, Paweł Pomastowski, Gulyaim N. Sagandykova, Bogusław Buszewski,

Firstly, the authors would like to thank the Editor and Reviewers for appreciation of our effort and secondly, for useful comments, remarks and valuable suggestions that led to the increasing of quality of the work. Therefore, the authors have addressed all the comments as explained below.

A revised manuscript has been submitted, in which the modified sections are highlighted in red. Thank the editor and reviewers again, who made great contributions to improve our paper.

Responses for Reviewer 1 comments:

Reviewer #1

The manuscript is now ready for publication.

Authors would like to thank the reviewer for this comment and appreciate time and efforts spent for review of this manuscript.

Reviewer 2 Report

In this revised version, the authors have clearly made a sincere effort to improve their manuscript. I only have a few relatively minor comments regarding the actual scientific content, and believe the work will be worthy of publication after these have been addressed.

Presentation-wise, while the English has been improved significantly in this version, there are still quite a few outright errors, particularly in the newly-added text, as well as parts that are grammatically correct, but difficult to read. In order to bring the readability of the paper up to the standard of a peer-reviewed paper, I would strongly recommend careful language editing.

Comments regarding scientific content:

Figure 1: please provide PDB accession codes for these structures

‘beta-LG has nine genetic variants’ – to my knowledge (and as stated in reference [4]) the known variants are A, B, C, D, E, F, G, H, I, J, and W, so eleven in total.

Table 1: The authors have clearly calculated the theoretical molecular weight of beta-LG from the gene sequence; however, the first 16 residues are a signal peptide that is cleaved off during processing. The mature protein has a molecular weight of 18281 Da, not 19883, explaining why 18 to 18.5 kDa was found experimentally.

Page 5, line 90: what is meant by ‘re-creasing’? Is this simply ‘refolding’ and a language issue, or am I missing something?

Page 5, line 108: ‘s-electron ions’ does not make sense. Replace with ‘ions of s-block elements’

Line 112-113: ‘the alpha-LA complex of zinc alpha-LA can be used’ - delete the second instance of ‘alpha-LA’

Line 116-117: as noted in the previous version, delete ‘or the identification method used’

Line 135: ‘acidic conditions (below 4) – specify that it is ‘pH below 4’

Figure 2 caption: ‘2A’, not ‘1A’

Table 4: the analytical method(s) for the 11th entry (reference [92]) seem to be missing

Section 2.3.1: The new text on ESI-MS is decent, but the English needs significant editing.

Line 456: Thank you for adding a reference on the statement that whey proteins form glutathione. It would be good to clarify that this is due to the high sulphur content of whey protein, as this is usually the rate-limiting step in GSH synthesis.

Language issues (listed here are the first 10 I noticed; there are more):

Page 2: ‘presented in Figure 1’

Figure 1 caption: ‘ferric atom’ sounds strange – use ‘iron atom’ instead

Page 2, line 41-44: this sentence goes on for too long to be understandable on the first read-through. Please split this into two sentences for readability.

Page 2, line 56: ‘dependent on posttranslational modifications’

Line 58: ‘by the precision of the analytical method’

Page 3: line 62-67: sentence is far too long

Line 67: ‘in a number of publications’

Page 5, line 123: ‘both lobes contain’

Page 5, line 98: ‘weakly folded’, not ‘weekly’ (i.e. once a week)

Line 125: ‘in each lobe, a single Fe atom…’ and ‘by amino acid side chains’

Author Response

Response to Reviewer 2 Comments

Ref. No.:  ijms-732358

Interactions of whey proteins with metal ions

Agnieszka Rodzik, Paweł Pomastowski, Gulyaim N. Sagandykova, Bogusław Buszewski,

Firstly, the authors would like to thank the Editor and Reviewers for appreciation of our effort and secondly, for useful comments, remarks and valuable suggestions that led to the increasing of quality of the work. Therefore, the authors have addressed all the comments as explained below.

A revised manuscript has been submitted, in which the modified sections are highlighted in red. Thank the editor and reviewers again, who made great contributions to improve our paper.

Responses for Reviewers 2 comments:

In this revised version, the authors have clearly made a sincere effort to improve their manuscript. I only have a few relatively minor comments regarding the actual scientific content, and believe the work will be worthy of publication after these have been addressed.

Presentation-wise, while the English has been improved significantly in this version, there are still quite a few outright errors, particularly in the newly-added text, as well as parts that are grammatically correct, but difficult to read. In order to bring the readability of the paper up to the standard of a peer-reviewed paper, I would strongly recommend careful language editing.

Comments regarding scientific content:

Point 1:  Figure 1: please provide PDB accession codes for these structures

Response 1: We would like to thank the reviewer  for the comment. PDB accession codes have been added according to Reviewer’s suggestions.

Point 2: ‘beta-LG has nine genetic variants’ – to my knowledge (and as stated in reference [4]) the known variants are A, B, C, D, E, F, G, H, I, J, and W, so eleven in total.

Response 2: We would like to thank the reviewer for this remark, which is certainly reasonable. The paragraph was revised with consideration of your comment.

Point 3: Table 1: The authors have clearly calculated the theoretical molecular weight of beta-LG from the gene sequence; however, the first 16 residues are a signal peptide that is cleaved off during processing. The mature protein has a molecular weight of 18281 Da, not 19883, explaining why 18 to 18.5 kDa was found experimentally.

Response 3: We would like to thank the reviewer for this remark. Authors suggest that reviewer may mention β-LG B consisting of 162 amino acids residues and according to Eigel et al. (Eigel W.N., Butler J.E., Ernstrom C.A., Farrell H.M., Harwalkar V.R., Jenness R., Whitney R., Nomenclature of Proteins of Cow's milk: Fifth Revision, J Dairy Sci 1984 67 1599-1631), its theoretical mass accounts for 18.277 kDa and this value has been added to the table. Unfortunately, authors could not find elsewhere the value mentioned by reviewer except some of the papers without citing the original publications reporting this value, although several papers mentioned the differences between the form consisting of 180 amino acids residues and that one of 162 without a signal peptide consisting from 18 amino acids residues (Folch M.J., Coll A., Sanchez A, Rapid Communication: Cloning and Sequencing of the cDNA Encoding Coat β-Lactoglobulin, J. Anim. Sci. 1993 71: 2832; Ali S., Clark A.J., Characterization of the gene encoding ovine beta-lactoglobulin, J. Mol. Biol. 1988 199 415-426). In addition, UniProt database entries have shown options such as 18.151 kDa (162 residues) and 18.263 kDa (163 residues). Authors hope that reviewer will be satisfied with work done.

Point 4: Page 5, line 90: what is meant by ‘re-creasing’? Is this simply ‘refolding’ and a language issue, or am I missing something?

Response 4: Authors would like to thank the reviewer for this comment. 'Re-creasing' has been changed to 'refolding'.

Point 5: Page 5, line 108: ‘s-electron ions’ does not make sense. Replace with ‘ions of s-block elements’

Response 5: Authors would like to thank the reviewer for this comment. This recommendation has been followed and 's-electron ions' was replaced to 'ions of s-block elements'.

Point 6: Line 112-113: ‘the alpha-LA complex of zinc alpha-LA can be used’ - delete the second instance of ‘alpha-LA’

Response 6: Authors would like to thank the reviewer for this comment. The second instance of ‘alpha-LA’ has been removed from mentioned sentence.

Point 7: Line 116-117: as noted in the previous version, delete ‘or the identification method used’

Response 7: Indeed, this point has been clarified during the previous revision, but has not been corrected on lines 116-117. Authors apologize for this mistake that has been corrected in the new revised version of the manuscript.

Point 8: Line 135: ‘acidic conditions (below 4) – specify that it is ‘pH below 4’

Response 8: This recommendation has been followed.

Point 9: Figure 2 caption: ‘2A’, not ‘1A’

Response 9: This mistake has been corrected.

Point 10: Table 4: the analytical method(s) for the 11th entry (reference [92]) seem to be missing

Response 10: Authors have added analytical methods for the mentioned entry in Table 4.

Point 11: Section 2.3.1: The new text on ESI-MS is decent, but the English needs significant editing.

Response 11: The manuscript has been passed through an extensive language check by native speaker and authors hope that at the moment english of the manuscript meets the high standards of the journal.

Point 12: Line 456: Thank you for adding a reference on the statement that whey proteins form glutathione. It would be good to clarify that this is due to the high sulphur content of whey protein, as this is usually the rate-limiting step in GSH synthesis.

Response 12: The Reviewer's suggestions have been taken into account and information about the sulphur content of whey protein has been added to the text.

Point 13: Language issues (listed here are the first 10 I noticed; there are more):

Page 2: ‘presented in Figure 1’

Figure 1 caption: ‘ferric atom’ sounds strange – use ‘iron atom’ instead

Page 2, line 41-44: this sentence goes on for too long to be understandable on the first read-through. Please split this into two sentences for readability.

Page 2, line 56: ‘dependent on posttranslational modifications’

Line 58: ‘by the precision of the analytical method’

Page 3: line 62-67: sentence is far too long

Line 67: ‘in a number of publications’

Page 5, line 123: ‘both lobes contain’

Page 5, line 98: ‘weakly folded’, not ‘weekly’ (i.e. once a week)

Line 125: ‘in each lobe, a single Fe atom…’ and ‘by amino acid side chains’

Response 13: Authors appreciate the reviewers recommendations and all of them have been followed in the revised version of the manuscript.